# Polyphenols in Urine and Cardiovascular Risk Factors: A Cross-Sectional Analysis Reveals Gender Differences in Spanish Adolescents from the SI! Program

**DOI:** 10.3390/antiox9100910

**Published:** 2020-09-24

**Authors:** Emily P. Laveriano-Santos, Isabella Parilli-Moser, Sonia L. Ramírez-Garza, Anna Tresserra-Rimbau, Carolina E. Storniolo, Ana María Ruiz-León, Ramón Estruch, Patricia Bodega, Mercedes de Miguel, Amaya de Cos-Gandoy, Vanesa Carral, Gloria Santos-Beneit, Juan M. Fernández-Alvira, Rodrigo Fernández-Jiménez, Valentín Fuster, Rosa M. Lamuela-Raventós

**Affiliations:** 1Department of Nutrition, Food Science and Gastronomy, Institute of Nutrition and Food Safety (INSA-UB), School of Pharmacy and Food Sciences XaRTA, University of Barcelona, 08921 Santa Coloma de Gramenet, Spain; emily.laveriano@gmail.com (E.P.L.-S.); iparillim@ub.edu (I.P.-M.); sonialrmz@gmail.com (S.L.R.-G.); carolinastorniolo@outlook.com (C.E.S.); 2Consorcio CIBER, M.P. Fisiopatología de la Obesidad y Nutrición (CIBERObn), Instituto de Salud Carlos III (ISCIII), 28029 Madrid, Spain; RESTRUCH@clinic.cat; 3Department of Internal Medicine, Hospital Clínic, Institutd’Investigacions Biomèdiques August Pi I Sunyer (IDIBAPS), University of Barcelona, 08036 Barcelona, Spain; amruiz@clinic.cat; 4Mediterranean Diet Foundation, 08021 Barcelona, Spain; 5Foundation for Science, Health and Education (SHE), 08008 Barcelona, Spain; pbodega@fundacionshe.org (P.B.); mdemiguel@fundacionshe.org (M.d.M.); adecos@fundacionshe.org (A.d.C.-G.); vcarral@fundacionshe.org (V.C.); gsantos@fundacionshe.org (G.S.-B.); 6Centro Nacional de Investigaciones Cardiovasculares Carlos III (F.S.P.), 28029 Madrid, Spain; jmfernandeza@cnic.es (J.M.F.-A.); rodrigo.fernandez@cnic.es (R.F.-J.); vfuster@cnic.es (V.F.); 7The Zena and Michael A. Wiener Cardiovascular Institute, Icahn School of Medicine at Mount Sinai, New York, NY 10029, USA; 8CIBER de Enfermedades Cardiovasculares (CIBERCV), 28029 Madrid, Spain; 9Hospital Universitario Clínico San Carlos, 28040 Madrid, Spain

**Keywords:** antioxidants, pediatric, body composition, cardiovascular, lipid profile, Folin–Ciocalteu

## Abstract

(1) Background: Epidemiological studies have shown an inverse association between polyphenol intake and cardiovascular risk factors (CVRFs) in adults, but few have provided information about adolescents. The aim of this study was to evaluate the relationship between urinary total polyphenol excretion (TPE) and CVRFs in adolescents. (2) Methods: A cross-sectional study was performed in 1194 Spanish adolescents from the SI! (*Salud Integral*) program. TPE in urine samples was determined by the Folin–Ciocalteu method, after solid-phase extraction, and categorized into quartiles. The association between TPE and CVRFs was estimated using mixed-effect linear regression and a structural equation model (SEM). (3) Results: Linear regression showed negative associations among the highest quartile of TPE and body fat percentage (B = −1.75, *p*-value = <0.001), triglycerides (TG) (B = −17.68, *p*-value = <0.001), total cholesterol (TC) (B = −8.66, *p*-value = 0.002), and low-density lipoprotein (LDL)-cholesterol (LDL-C) (B = −4.09, *p*-value = 0.008) in boys, after adjusting for all confounder variables. Negative associations between TPE quartiles and systolic blood pressure (SBP), diastolic blood pressure (DBP), and TC were also found in girls. Moreover, a structural equation model revealed that TPE was directly associated with body composition and blood glucose and indirectly associated with blood pressure, TG, LDL-C, and high-density lipoprotein-cholesterol (HDL-C) in boys. (4) Conclusions: Higher concentrations of TPE were associated with a better profile of cardiovascular health, especially in boys, while in girls, the association was not as strong.

## 1. Introduction

Although the clinical burden of cardiovascular disease (CVD) mainly occurs in adulthood, the process of developing CVD begins early in life and progresses throughout the lifespan. The principal cardiovascular risk factors (CVRFs) are obesity, diabetes mellitus, and hypertension, which are normally related to modifiable lifestyle factors, such as an unhealthy diet, physical inactivity, smoking, and excessive alcohol intake [1]. The high prevalence of obesity in the current adolescent population is associated with unhealthy habits, such as an inadequate diet and physical inactivity. This is of particular concern because the excess weight in childhood and adolescence is directly associated with hypertension, an adverse lipid profile, type II diabetes, and early atherosclerotic lesions, which can increase the risk of developing CVD during adulthood [2].

According to preliminary studies, the Mediterranean diet plays an important role in the prevention of CVRFs, such as diabetes, obesity, and hypertension [3,4,5,6]. Most evidence for the positive cardiovascular health effects of the Mediterranean diet indicates that bioactive compounds, including polyphenols, are, in part, responsible [3]. The intake of polyphenol-rich foods, such as vegetables, fruits, olive oil, and seeds, has been inversely associated with cardiovascular risk factors in elderly populations [6,7]. The beneficial health effects of polyphenols depend on intake and bioavailability, which varies from one molecule to another, and among individuals [8].

Most epidemiological studies have determined polyphenol intake using traditional dietary assessment tools, such as food frequency questionnaires or 24-h diet recall. A less biased and potentially more accurate approach is the determination of urinary polyphenols by the Folin–Ciocalteu assay, which can serve as a biomarker of total polyphenol intake and fruit and vegetable consumption [9,10].

The relationship of polyphenol biomarkers with CVD or CVRFs has been observed in several studies, suggesting that it is of great importance to maintain polyphenol biomarkers at high levels [11]. Lower all-cause and CVD mortality risks have been observed at higher total urinary polyphenol levels, especially enterolignans concentrations (enterolactone). In addition, urine excretion of enterolignans has been inversely associated with C-reactive protein (CRP) levels and metabolic syndrome components, such as type 2 diabetes (T2D) and obesity. Metabolites from flavanones (naringenin and hesperetin), flavonols (quercetin and isorhamnetin), phenolic acids (caffeic acid), and enterolignans (enterolactone) in spot urine samples have been significantly associated with a lower T2D risk [12]. Moreover, levels of urinary polyphenol biomarkers (caffeic acid, ferulic acid, 3-hydroxybenzoic acid), and especially gut microbial metabolites of polyphenols, have been inversely associated with overweight and obesity. This negative association has been more pronounced in the participants with higher CRP levels, a marker of chronic inflammation and a predictor of all-cause cardiovascular mortality [13].

The aim of the present study was to determine the association between total polyphenol excretion (TPE) in urine, as a biomarker of total polyphenol intake, and CVRFs in adolescents from the SI! (*Salud Integral*) program.

## 2. Materials and Methods

### 2.1. Study Design and Participants

The SI! (*Salud Integral*) program for secondary schools is a well-established cluster-randomized controlled intervention trial, registered at ClinicalTrials.gov (NCT03504059), aimed at evaluating the effectiveness of an educational intervention to promote cardiovascular health in 1326 adolescents from 24 secondary public schools in Madrid and Barcelona, Spain. Schools were randomized 1:1:1 to receive a short-term (2-year) or a long-term (4-year) comprehensive educational program or the usual curriculum (control). The primary outcome was a change in obesity and other cardiovascular health parameters. The full details of the design of the SI! Program for Secondary Schools intervention have been published elsewhere [14].

The study protocol and procedures were approved by the Ethical Committee of *the Instituto de Salud Carlos III* in Madrid (CEI PI 35_2016), the *Fundació Unió Catalana d’Hospitals* (CEI 16/41), and the University of Barcelona (IRB00003099). Written informed consent was obtained from the parents or the legal guardians of all the participants. 

The inclusion criteria were availability of spot urine samples, no diagnosis of diabetes or hypertension, and not having taken any drugs or supplements the day prior to the data collection at baseline. Based on these criteria, 1194 subjects were included in the analysis (Figure 1).

### 2.2. Anthropometry and Body Composition

Weight was measured to the nearest 0.1 kg using a digital scale (OMRON BF511, OMRON HEALTHCARE Co., Muko, Kyoto, Japan), and height to the nearest 0.1 cm with a portable stadiometer (SECA 213 of 0.1 cm precision), with participants wearing light clothes and no shoes. The body mass index (BMI) was calculated as weight in kilograms divided by square height in meters (kg/m^2^). The body fat percentage was estimated by bioelectrical impedance using a tetrapolar OMRON BF511. Waist circumference (WC) was measured in triplicate to the nearest 0.1 cm using a non-stretchable Holtain tape [14]. The waist-to-height ratio (WtHR) was calculated as WC (cm) divided by height (cm). Measurements were taken early in the morning after overnight fasting. BMI and WC z-scores were calculated according to the age- and gender-specific median of the International Obesity Group (IOTF) and the National Health and Nutrition Examination Survey (NHANES), respectively [15,16].

### 2.3. Blood Pressure

Blood pressure (BP) was measured in the sitting position using a digital monitor OMRON M6 (OMRON HEALTHCARE Co., Muko, Kyoto, Japan). Duplicate measurements were taken at two- or three-minute intervals after the participants relaxed. If these differed by more than 10 mmHg for systolic blood pressure (SBP) and/or more than 5 mmHg for diastolic blood pressure (DBP), a third measurement was taken. Average values were calculated for the final SBP and DBP [14].

SBP and DBP z-scores were calculated according to the High Blood Pressure Working Group of the National Blood Pressure Education Program for children and adolescents [17].

### 2.4. Blood Analyte Measurements

Glucose and lipid profile (total cholesterol (TC), high-density lipoprotein-cholesterol (HDL-C), low-density lipoprotein-cholesterol (LDL-C), and triglycerides (TG)) levels were analyzed using a portable CardioCheck^®^ Plus (PTS Diagnostic, Indianapolis, IN, USA) biochemical analyzer in finger-prick capillary whole blood samples (approximately 40 µL) taken early in the morning after overnight fasting. The coefficients of variation (CV) of TC, HDL-C, and glucose were 4.9%, 8.7%, and 4.0%, respectively [18].

### 2.5. TPE Determination in Urine

Fasting spot urine samples were collected in the morning and were immediately stored at −80 °C until analysis [14]. The validated Folin-Ciocalteu (F-C) spectrophotometric method used to determine TPE concentrations in urine has been previously described by Medina-Remón et al. This method includes a previous solid-phase extraction using 96-well plate hydrophilic-lipophilic-balanced cartridges water-wettable and reversed-phase solvent (Oasis^®^ MAX 30 mg, Waters, Milford, MA, USA) to remove interferences with the F-C reagent (Sigma-Aldrich, St. Louis, MO, USA). Gallic acid (Sigma-Aldrich, St. Louis, MO, USA) was used for curve calibration. The method has a percent relative standard deviation (precision) below 7.2% in each concentration from curve calibration. Recovery values were between 82.5 and 105.7%. The limit of detection (LOD) for gallic acid equivalent (GAE) was 0.07 mg/L [9]. Creatinine in urine samples was measured following the adapted Jaffé alkaline picrate method for thermo microtiter 96-well plates, according to Medina-Remón et al. [9]. In epidemiological studies and in the absence of disease, creatinine concentrations in urine can be used to determine urinary excretion of compounds in spot urine samples [9,19,20]. We considered values of mg of GAE and creatinine if the CV between measures were less than 15%. Finally, TPE was expressed as mg GAE/g creatinine. 

### 2.6. Covariate Assessment

Information about dietary intake was assessed using an updated version of the validated 157-items semi-quantitative food frequency questionnaire (FFQ) [21]. From standard portions and frequencies of consumption, all items were calculated and reported in g per day (g/d) using Spanish food composition tables [22,23]. Dietary intake includes total energy, protein, carbohydrates, fiber, total fat, saturated fatty acids (SFA), monounsaturated fatty acids (MUFAs), and polyunsaturated fatty acids (PUFAs). Food groups intake includes vegetables, fruits, legumes, cereals, dairy products, meat, olive oil, fish, nut, cookies, pastries, and sweets, chocolate, and soft drinks. Nutrient and food groups’ intake was adjusted for total energy intake using the residual method [24].

A self-completed questionnaire for parents or legal guardians was used to assess sociodemographic parameters (place of birth, education level, and house income) [14]. The place of birth was categorized according to the area in which the participants were born, like Spain, the rest of Europe, Latin America, Africa, and others. The parental education level was categorized as low (families without studies or with primary studies), medium (secondary studies and professional training), and high (university studies), according to the International Standard Classification of Education (ISCED) [25]. As in the study by Bodega et al. [26], the highest parental education level was considered as a covariate in this study. House income was categorized into three levels (low, medium, and high) based on the annual survey of salary for the Spanish population [27].

Physical activity was measured with a triaxial Actigraph wGT3X-BT accelerometer (ActiGraph Corporation, Pensacola, FL, USA) on the non-dominant wrist for 7 consecutive days (except when bathing or swimming) [14]. The cut-off points of Chandler (2016) were applied to estimate total activity and minutes spent in moderate-to-vigorous physical activity [28].

Puberty development was assessed using pictograms, according to Tanner maturation stages [29].

### 2.7. Statistical Analysis

The baseline characteristics of participants were presented as means and standard deviations (SD) for continuous variables and frequencies and percentages for categorical variables. To assess the relation between TPE and CVRFs, participants were categorized into quartiles of TPE (mg GAE/g creatinine): Q1 (<71.8), Q2 (71.9–111.1), Q3 (111.2–161.2), and Q4 (>161.2). Continuous variables were used to compare the unadjusted sample means across TPE quartiles by one-way analysis of variance (ANOVA). Chi-square test analysis was used to assess qualitative variables across TPE quartiles. 

Mixed-effect linear regression models were considered to evaluate associations between TPE (continuous and quartiles) and CVRFs (body composition, BP, glucose, TG, TC, HDL-C, and LDL-C), adjusted for factors previously related to TPE and CVRFs. The first model was unadjusted. In the second model, TPE was adjusted for gender (only for total participants), age (continuous), fasting (yes/no), moderate-to-vigorous physical activity (quartiles), Tanner stage (from I to V), a high parental education level (yes/no), place of birth (Spain, Rest of Europe, Latin America, Africa, others), and household income (low, medium, and high). The third model was additionally adjusted for energy intake (quartiles), fiber (quartiles of energy-adjusted intake), total fat (quartiles of energy-adjusted intake), MUFAs (quartiles of energy-adjusted intake), PUFAs (quartiles of energy-adjusted intake). The BMI and WtHR were also considered in the third model regression between TPE and blood analytes (glucose, TG, TC, HDL-C, and LDL-C). Municipalities (Barcelona/Madrid) were included as a random effect. To accommodate the use of some categorical variables, we estimated parameters using weighted least squares with robust standard errors. We evaluated potential effect modification in the association between quartiles of TPE and CVRFs by gender in an interaction analysis using the cross-product term between TPE and gender. This analysis was also stratified by gender to evaluate potential modification. Linear trends between TPE and mean of each CVRF were assessed using orthogonal polynomial contrasts.

Based on linear regression models, the data was re-analyzed using structural equation modeling (SEM) with robust maximum likelihood estimation to examine the relationship between TPE (continuous) and CVRFs (continuous). Our hypothesized model of study is shown in Figure 2. The main dependent variables were TG, TC, LDL-C as observed variables; body composition (included the WC z-score, the BMI z-score, and body fat percentage) and blood pressure (BP) (included the SBP z-score and the DBP z-score) as latent variables that are not measured directly. TPE was considered an independent variable. Other variables observed were considered as covariates: age, gender, physical activity, fasting, Tanner scale, energy intake, fat intake, MUFAs, PUFAs, fiber, high parental education, place of birth, and household income. The goodness-of-fit index model for SEM included the standardized root mean square residual (SRMR < 0.08) [30].

All statistical analyses were conducted using the Stata statistical software package version 16.0 (StataCorp, College Station, TX, USA). Statistical tests were two-sided, and *p*-values below 0.05 were considered significant.

## 3. Results

### 3.1. General Characteristics

Table 1 summarizes the baseline gender-stratified characteristics of the 1194 participants (90% of the cohort) who were included in this cross-sectional study. Nearly half (48%) were girls with a mean age of 12.0 ± 0.5 years. The mean TPE was 125.7 ± 76.8 mg GAE/g creatinine, and the median 111.2 mg GAE/g creatinine with a CV of 61%, the minimum value was 5.1 mg GAE/g creatinine and the maximum 534.3 mg GAE/g creatinine. For the concentration of creatinine, the mean was 1.27 ± 0.49 g/L, with a CV of 39%. No significant differences were found in the TPE, according to gender. Regarding the CVRFs, significantly higher mean values of the BMI z-score, WC, the WC-z score, WtHR, SBP, and glucose were observed in boys compared to girls. Boys were also more sedentary, although they walked more than girls. On the other hand, significantly higher values of body fat percentage, DBP, the DBP-z score, TG, and TC were reported in girls. We observed gender-related differences in dietary food intake: boys showed a trend of having higher intakes of energy and less fiber intake.

The general characteristics of the participants, according to urinary TPE quartiles, are shown in Table 2. Significant differences in the mean BMI (*p*-trend = 0.017), the BMI z-score (*p*-trend = 0.022), body fat (*p*-trend = 0.003), glucose (*p*-trend = 0.009), TG (*p*-trend = 0.017), and TC (*p*-trend = 0.002) were observed among the quartiles. Differences in TPE stratified by gender are shown in Table A1. Lower values of body fat (*p*-trend = 0.039), glucose (*p*-trend = 0.006), TG (*p*-trend = 0.011), TC (*p*-trend = 0.012), and LDL-C (*p*-value = 0.045) were observed between quartiles of TPE in adolescent boys. The results indicated that boys with the highest TPE had a better CVRF profile, but this did not apply to girls. Other factors with significant differences among quartiles only in boys were the level of maternal education and place of birth. Significant differences were only found among quartiles regarding municipality in girls.

We summarized dietary nutrients and the main food groups intake, according to quartiles of TPE, in Table 3. We did not observe significant differences between quartile groups except for cookies, pastries, sweets, and snacks, the intake of which tended to decrease inversely with quartiles, and legumes that presented higher values in the first and the third quartile, although no significant trend was appreciated. 

### 3.2. TPE, Body Composition, and BP

The associations among TPE quartiles with body composition and BP are shown in Table 4. After adjustment for age, gender (only for total participants), physical activity, fasting, Tanner scale, energy intake, fat intake, MUFAs, PUFAs, fiber, high parental education, place of birth, and house income in the third model, the highest quartile of TPE was negatively associated with body fat percentage (B= −1.75, *p*-value < 0.001, 95% confidence interval (CI = −2.16; −1.36)), compared to the lowest quartile of TPE for boys. On the other hand, in girls, the highest quartile of TPE was negatively associated with the SBP z-score and the DBP z-score, compared to the lowest quartile of TPE. 

### 3.3. TPE and Blood Analytes

Table 5 shows the relationship between urinary TPE and blood analysis estimated using linear regression. For all the participants, negative associations were found among the highest quartile of urinary polyphenols and TG (B= −9.31, *p*-value < 0.001, 95% CI= −12.69; −5.15), TC (B = −7.09, *p*-value < 0.001, 95% CI = −9.28; −4.98), and LDL-C (B= −1.98, *p*-value = 0.006, 95% CI = −4.09; −0.11), compared to the lowest quartile of TPE, after full adjustment for potential confounders. In the gender-stratified analysis, a negative association was found only in boys among the highest quartile of TPE and TG, and TC and LDL-C, compared to the lowest quartile in all regression models, with the exception of the second model in TG in which no significant result was found. No association was observed in boys between the highest and lower quartile of TPE and HDL-C in the fully adjusted model. In the case of the girls, a negative association was found between the highest quartile of TPE and TC, after adjusting for all confounder variables. Although the association between TPE quartiles and LDL-C and HDL-C was not significant in girls, an inverse trend was observed in the fully adjusted regression model.

### 3.4. TPE and CVRFs

Figure 2 illustrates our hypothesized relationship model between TPE and CVRFs, using SEM in a total of 566 participants. The model fit indicated a good fit in which the SRMR was 0.076, after adjustment for age, gender, physical activity, fasting, Tanner scale, energy, total fat, MUFAs, PUFAs, fiber intake, high parental education, place of birth, and house income. As expected, TPE was directly and negatively associated with body composition (B = −0.02, *p*-value = 0.020, 95% CI = −0.03; −0) and TC (B = −0.04, *p*-value = 0.008, 95% CI = −0.08; −0.01). Moreover, indirect and negative associations between TPE and TG (B = −0.02, *p*-value = 0.020, 95% CI = −0.03; −0) and LDL-C (B = −0.01, *p*-value = 0.044, 95% CI = −0.01; −0) were found. TPE was also indirectly and positively associated with HDL-C (B = 0.01, *p*-value = 0.018, 95% CI = 0; 0.01). No association was observed between TPE and BP (SBP z-score and DBP z-score). In addition, we observed that body composition was positively and directly associated with BP, blood glucose, and LDL-C and negatively associated with HDL-C. 

In the boys model, a direct and negative relationship was observed among TPE with body composition (B = −0.03, *p*-value = 0.009, 95% CI = −0.05; −0.01) and blood glucose (B = −0.02, *p*-value = 0.022, 95% CI = −0.03; −0) using SEM. Indirect and negative associations were found among TPE with BP (B = −0.001, *p*-value = 0.003, 95% CI = −0; −0), TG (B = −0.03, *p*-value = 0.009, 95% CI = −0.05; −0.01), LDL-C ((B = −0.02, *p*-value = 0.013, 95% CI = −0.03; −0). TPE and HDL-C (B = 0.01, *p*-value = 0.008, 95% CI = 0; 0.02) were indirectly and positively associated. Although there was no association between TPE and CVRFs in girls, direct associations were found between body composition and TG with BP (Figure A1).

## 4. Discussion

To our knowledge, this is the first study to assess the association between total polyphenols in urine and CVRFs in adolescents. In this baseline cross-sectional study, we observed that a higher concentration of TPE was associated with a better profile of CVRFs, even though we observed different results according to gender. 

The inverse association observed between TPE and body composition has been shown in a previous study by our research group in the PREDIMED (Prevención con Dieta Mediterránea) cohort, in which higher TPE in urine was associated with lower values of body weight, BMI, WC, and WtHR in elderly participants at high cardiovascular risk [31]. Although in this study, a direct and negative association between TPE and variables related to body composition (BMI z-score, WC z-score, and body fat percentage) was found in total participants using SEM, these results were gender-dependent, and stratified results showed significant associations only in boys. This result could be partly attributed to dietary intake. However, in the SEM and regression models, energy intake, fiber, total fat, MUFAs, and PUFAs intake were considered as covariates for removing the possible effects of the diet. This finding was consistent with a recent study, which reported that adolescents with a high total polyphenol intake presented lower BMI z-score values, even though it was based on an FFQ, in contrast with our work [32].

No relationship between urinary polyphenols and BP was observed in the total participants. Although, in linear regression models, we observed a negative association between TPE and SBP z-score and DBP z-score in girls, after SEM analysis, we found that body composition mediated this relationship. Besides, we observed indirect and a negative association between TPE and BP in boys after SEM analysis. The present finding was consistent with previous studies. A cross-sectional study found an inverse association between TPE and BP in subjects at high cardiovascular risk [33]. In the PREDIMED study, a diet rich in polyphenols was found to reduce SBP and DBP in adults with hypertension, possibly by stimulating the formation of vasoprotective factors, such as nitric oxide in plasma [6]. However, in a similar trial, Guo et al. only observed this correlation with SBP after five years of the intervention [34]. The difference between the results could be due to the different health conditions of the participants: in our case, young individuals were mainly normotensive, as opposed to a hypertensive elderly population. 

A higher TPE in boys was associated with lower TG, TC, and LDL-C values but not HDL-C in the fully adjusted linear regression model. However, in the SEM analysis, we observed an indirect and positive association between TPE and HDL, mediated by body composition. In line with these findings, results from the PREDIMED cohort showed a similar inverse association with TG levels, but not with TC and LDL-C [34]. Previous studies reporting polyphenol intake in adults using questionnaires found that TG values improved as the polyphenol intake increased. A study of Polish adults indicated that TG values were significantly lower among individuals in the highest quartile of polyphenol intake [35,36]. The effects of polyphenols on lipids are not clear due to several factors: the little knowledge of active metabolites, the inter- and intra-individual variability of the intestinal microflora, and the number of subjects included in the studies. Nevertheless, possible mechanisms that could explain the favorable association between polyphenols and the lipid profile have been mentioned, such as a reduction in lipogenesis; an increase in lipolysis; stimulation of fatty acid β-oxidation; inhibition of adipocyte differentiation and growth; inhibition of expression and secretion of pro-inflammatory molecules; a decrease in oxidative stress, and an increase in antioxidant capacity in adipose tissue [37]. Polyphenols can decrease lipid digestion and absorption by reducing the activities of digestive enzymes and lipid emulsification. Green tea catechins, resveratrol, and curcumin have been considered to decrease fat accumulation in adipocytes by activating adenosine monophosphate-activated protein kinase (AMPK) and down-regulating the expression of lipogenic genes. These polyphenols have also been seen to increase lipolysis and stimulate fatty acid β-oxidation by upregulating hormone-sensitive lipase [38].

In boys, SEM analysis showed a direct and negative association between TPE and blood glucose. In a previous clinical trial, a high intake of polyphenol-rich foods for 8 weeks reduced glucose concentrations in plasma and increased insulin secretion in adults at high cardiovascular risk [39]. Moreover, in the PREDIMED trial, a higher intake of total polyphenols, total flavonoids, stilbenes, and some flavonoid subclasses was inversely and linearly associated with the incidence of type 2 diabetes [40], and in the PREDIMED-Plus trial, high intakes of some polyphenols were inversely associated with the prevalence of type 2 diabetes in adults with metabolic syndrome, particularly in overweight subjects [41].

In this study, we observed gender-related differences between TPE and CVRFs. Although TPE concentrations are similar in boys and girls, inter-individual variability could be present. The heterogeneity in the cardiometabolic response to polyphenols may be influenced by gender, age, health status, and medication, variables that were considered in our regression models [42]. Sex-gender differences could have important implications in the bioavailability, distribution, metabolism, and excretion of phenolic compounds [43]. Sexual dimorphism in the absorption of polyphenols has been explained in animal models, where the expression of uridine 5’-diphospho (UDP)-glucuronosyltransferases (UGTs), the enzyme responsible for the absorption of polyphenols in the small intestine, was higher in males than females, affecting the glucuronidation of polyphenols [44]. In the human liver, some polyphenols, like caffeic acid, tyrosol, genistein, and daidzein, are metabolized by cytochrome P450 (CYPs) [45,46]. It is well known that CYP2B6, CYP2A6, and CYP3A activity is up-regulated by estrogen levels, indicating a higher activity in women than men [45,47]. Finally, gender-difference influences the excretion of polyphenols in the urine. In adults with hypercholesterolemia, men excreted more concentration of 3,5-diOH-benzoic acid, t-coutaric acid, naringenin, 4-hydroxybenzoic acid, and 4-hydroxyphenyl acetic acid than women [48]. Similar results were found in older adults, where men excreted more polyphenols than women [49]. Another important factor of this variability is the composition of gut microbiota; microbiota participates in the metabolism of polyphenols that cannot be absorbed in the small intestine. Microbial phenolic compounds might have a higher impact on human health than their parental polyphenols [50]. Although microbiota contributes to inter-individual variability, in this study, we did not evaluate microbial compounds.

In addition, there are gender differences in adiposity, metabolism, and predisposition for metabolic dysfunction that can be partially driven by sex hormones. Some studies have analyzed the effects of sex hormone levels on plasma lipid levels in children, and differences in HDL-C levels by sex during puberty have been related to the rise of testosterone levels in boys [51]. In addition, animal models have demonstrated that sex and sex hormones influence adipose tissue, gene expression profiles, regulating insulin resistance and lipolysis, as well as inflammatory tone and obesity [52]. Depending on sex, the level of specific sex hormones can improve or worsen metabolic dysfunction. Estrogen generally provides a protective effect in females, while adequate androgen levels in males are important in promoting appropriate adiposity and metabolic status, and an increase of testosterone decreases abdominal obesity and the metabolic risk profile [52]. Indeed, a study that included healthy schoolchildren ranging from 12 to 15 years old showed that sex hormone-binding globulin levels were related to a decrease in HDL-C and apolipoprotein A-I levels during puberty in boys and to a decrease in TG levels during puberty in both sexes [51]. Nevertheless, more mechanistic studies are needed to fully understand sex dimorphism.

A strong point of our study is the use of TPE as a biomarker to determine polyphenol intake, as this is a more reliable and accurate approach than food questionnaires. Moreover, the F-C method with microplates was used, which is a fast and cheap and reliable method to determine TPE [9]. 

The principal limitation of this study is the cross-sectional analysis design, and causality associations between exposure and outcomes cannot be established. Another limitation is that we did not measure specific phenolic metabolites but only total polyphenol excretion in urine. Moreover, we cannot rule out the possibility of residual confounding of the associations observed. In addition, the limited number of publications evaluating the relationship between polyphenols and cardiovascular health in adolescents did not allow comparison of the results with previous studies in similar populations.

## 5. Conclusions

In summary, the results of this study suggest an association between TPE and better CVRFs, mainly in male adolescents. However, direct associations between body composition and TG with BP were also found in females.

## Figures and Tables

**Figure 1 antioxidants-09-00910-f001:**
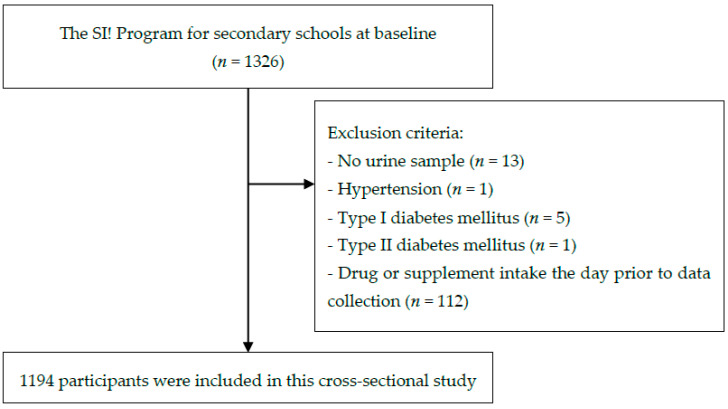
Flow chart of participant selection.

**Figure 2 antioxidants-09-00910-f002:**
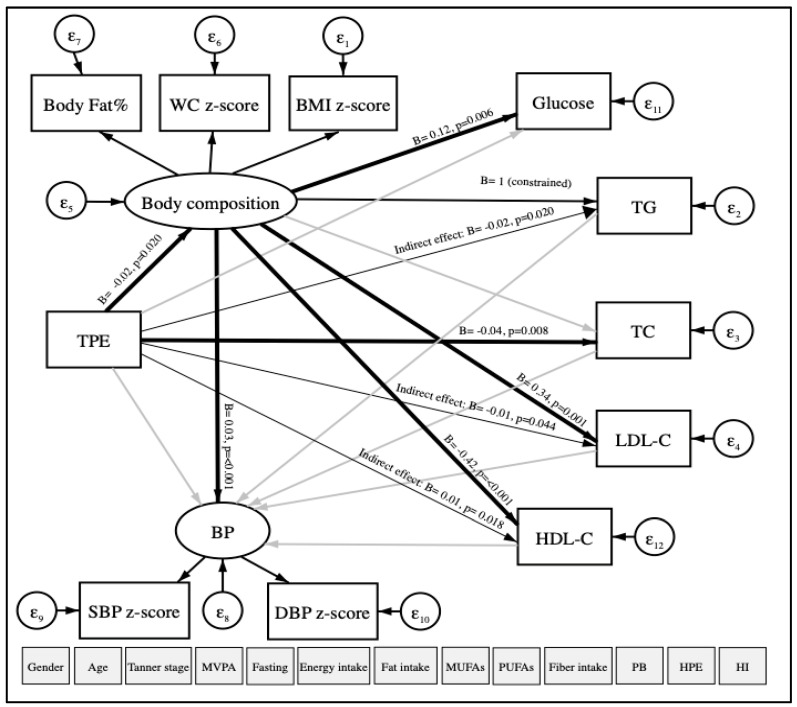
Path diagram of the association between TPE and CVRFs using structural equation modeling. TPE: Total polyphenol excretion in urine. CVRFs: Cardiovascular risk factors. WC: Waist circumference. BMI: Body mass index. BP: Blood pressure. SBP: Systolic blood pressure. DBP: Diastolic blood pressure. TG: Triglycerides. TC: Total cholesterol. LDL-C: Low-density lipoprotein-cholesterol. HDL-C: High-density lipoprotein-cholesterol. MVPA: Moderate-to-vigorous physical activity. MUFAs: Monounsaturated fatty acids. PUFAs: Polyunsaturated fatty acids. PB: Place of birth. HPE: High parental education. HI: Household income. ε: error. Oval circles indicate a latent variable that is not measured directly. Age, gender, MVPA, fasting, Tanner scale, energy intake, fat intake, MUFAs, PUFAs, fiber intake, PB, HPE, and HI were considered as covariates. Significant paths (*p* < 0.05) are shown as black arrows, and non-significant paths are shown as grey arrows. Direct associations are presented with wider arrow widths than the indirect associations. Unstandardized regression coefficients are at the end of each arrow.

**Table 1 antioxidants-09-00910-t001:** Baseline characteristics of the study population, according to gender (*n* = 1194).

Variable	Total	Girls	Boys	*p*-Value
		*n* = 569 (48%)	*n* = 625 (52%)	
Age (y), mean (SD)	12.04 (0.46)	11.99 (0.41)	12.08 (0.49)	**0.002**
Body composition, mean (SD)				
BMI (kg/m^2^)	20.16 (3.78)	20.04 (3.71)	20.26 (3.83)	0.303
BMI z-score	0.65 (1.06)	0.54 (1.03)	0.74 (1.08)	**<0.001**
WC (cm)	71.87 (10.24)	70.47 (8.99)	73.14 (11.19)	**<0.001**
WC z-score	0.39 (0.86)	0.29 (0.71)	0.47 (0.91)	**<0.001**
WtHR	0.46 (0.06)	0.45 (0.06)	0.47 (0.07)	**<0.001**
Body fat (%)	22.93 (8.31)	24.45 (7.93)	21.52 (8.41)	**<0.001**
Blood pressure, mean (SD)				
SBP (mmHg)	104.59 (10.71)	103.47 (10.19)	105.60 (11.07)	**<0.001**
SBP z-score	−0.30 (0.95)	−0.35 (0.94)	−0.26 (0.96)	0.099
DBP (mmHg)	61.71 (8.74)	62.68 (8.58)	60.83 (8.81)	**<0.001**
DBP z-score	−0.12 (0.76)	−0.07 (0.78)	−0.17 (0.75)	**0.019**
Blood analytes, mean (SD)				
Glucose (mg/dL)	102.89 (11.76)	101.76 (11.70)	103.91 (11.72)	**0.002**
TG (mg/dL)	76.55 (41.07)	80.11 (37.79)	73.35 (43.59)	**0.005**
TC (mg/dL)	152.63 (34.46)	154.39 (32.54)	151.05 (36.04)	0.099
LDL-C (mg/dL)	77.96 (25.78)	78.41 (24.74)	77.47 (26.86)	0.585
HDL-C (mg/dL)	62.85 (15.89)	62.78 (14.26)	62.91 (17.24)	0.891
Physical activity, mean (SD)				
Sedentary (min/day)	605.89 (68.52)	596.78 (65.53)	614.51 (70.21)	**<0.001**
MVPA (min/day)	74.61 (23.41)	77.17 (22.89)	72.19 (23.66)	**<0.001**
Step counts per day	12,104.48 (2419.28)	11,711.49 (2175.84)	12,476.33 (2576.14)	**<0.001**
Tanner scale, mean (SD)	3.16 (0.85)	3.16 (0.84)	3.17 (0.85)	0.745
Dietary intake, mean (SD)				
Total energy (kcal/d)	2435.27 (546.59)	2343.71 (512.62)	2518.21 (563.47)	**<0.001**
Protein (g/d)	116.46 (16.65)	115.61 (15.98)	117.22 (17.22)	0.158
Carbohydrates (g/d)	243.83 (35.48)	244.13 (35.23)	243.55 (35.74)	0.812
Fiber (g/d)	27.91 (7.01)	28.62 (6.95)	27.27 (7.00)	**0.005**
Total fat (g/d)	110.31 (14.13)	110.52 (13.79)	110.11 (14.45)	0.672
SFAs (g/d)	35.89 (5.67)	35.98 (5.45)	35.80 (5.86)	0.642
MUFAs (g/d)	49.96 (8.98)	49.79 (9.66)	50.11 (8.33)	0.613
PUFAs (g/d)	18.70 (3.99)	18.66 (4.12)	18.74 (3.88)	0.780
Creatinine (g/L)	1.27 (0.49)	1.35 (0.51)	1.19 (0.45)	**<0.001**
TPE (mg GAE/g creatinine), mean (SD)	125.72 (76.84)	122.41 (74.59)	128.73 (78.78)	0.156
Place of birth, *n* (%)				0.800
Spain	1082 (90.62)	520 (91.39)	562 (89.92)	
Rest of Europe	15 (1.26)	7 (1.23)	8 (1.28)	
Latin America	58 (4.86)	26 (4.57)	32 (5.12)	
Africa	10 (0.84)	3 (0.53)	7 (1.12)	
Other	29 (2.43)	13 (2.28)	16 (2.56)	
Education of mother, *n* (%)				0.113
Low	282 (26.09)	152 (28.95)	130 (23.38)	
Medium	429 (39.69)	201 (38.29)	228 (41.01)	
High	370 (34.23)	172 (32.76)	198 (35.61)	
Education of father, *n* (%)				0.958
Low	248 (30.35)	115 (29.95)	133 (30.72)	
Medium	330 (40.39)	155 (40.36)	175 (40.42)	
High	239 (29.25)	114 (29.69)	125 (28.87)	
Household income, *n* (%)				0.748
Low	366 (33.09)	181 (34.09)	185 (32.17)	
Medium	348 (31.46)	167 (31.45)	181 (31.48)	
High	392 (35.44)	183 (34.46)	209 (36.35)	
Municipality, *n* (%)				0.351
Barcelona	845 (70.77)	410 (72.06)	435 (69.60)	
Madrid	349 (29.23)	159 (27.94)	190 (30.40)	

TPE: Total polyphenol excretion in urine. GAE: Gallic acid equivalent. BMI: Body mass index. WC: Waist circumference. WtHR: Waist-to-height ratio. SBP: Systolic blood pressure. DBP: Diastolic blood pressure. TG: Triglycerides. TC: Total cholesterol. LDL-C: Low-density lipoprotein-cholesterol. HDL-C: High-density lipoprotein-cholesterol. MVPA: Moderate-to-vigorous physical activity. SFAs: Saturated fatty acids. MUFAs: Monounsaturated fatty acids. PUFAs: Polyunsaturated fatty acids. Statistical analyses were undertaken using the t-test for continuous variables and the chi-square (χ2) test for categorical variables. *p*-values refer to differences between girls and boy values < 0.05; values shown in bold are statistically significant.

**Table 2 antioxidants-09-00910-t002:** Baseline characteristics of the study population, according to TPE quartiles (mg GAE in urine/g creatinine).

Variable		TPE (mg GAE in urine/g creatinine)					
	*n*	Q1 (<71.8)	Q2 (71.9–111.1)	Q3 (111.2–161.2)	Q4 (>161.2)	*p-*Value	*p-*Trend
Number of participants	1194	299	298	299	298		
Gender (Girls), *n* (%)	569	139 (46.49)	158 (53.02)	141 (47.16)	131 (43.96)	0.154	0.300
Age (y), mean (SD)	1194	12.03 (0.45)	12.09 (0.50)	12.01 (0.43)	12.02 (0.46)	0.123	0.341
Body composition, mean (SD)							
BMI (kg/m^2^)	1193	20.44 (4.02)	20.47 (3.86)	19.85 (3.34)	19.86 (3.85)	0.057	**0.017**
BMI z-score	1193	0.72 (1.10)	0.72 (1.00)	0.59 (1.00)	0.55 (1.10)	0.115	**0.022**
WC (cm)	1194	72.49 (10.81)	72.59 (10.46)	70.80 (9.30)	71.60 (10.29)	0.110	0.093
WC z-score	1194	0.43 (0.87)	0.44 (0.84)	0.31 (0.86)	0.36 (0.87)	0.211	0.140
WtHR	1194	0.47 (0.07)	0.47 (0.07)	0.46 (0.06)	0.46 (0.06)	0.195	0.085
Body fat (%)	1182	23.79 (8.57)	23.57 (8.26)	22.25 (8.09)	22.10 (8.22)	**0.019**	**0.003**
Blood pressure, mean (SD)							
SBP (mmHg)	1191	105.35 (10.66)	104.35 (10.92)	104.24 (10.30)	104.40 (10.98)	0.559	0.289
SBP z-score	1191	−0.23 (0.95)	−0.34 (0.97)	−0.32 (0.92)	−0.32 (0.96)	0.436	0.274
DBP (mmHg)	1191	62.34 (9.19)	60.94 (8.56)	61.22 (8.54)	62.35 (8.61)	0.095	0.891
DBP z-score	1191	−0.06 (0.80)	−0.20 (0.75)	−0.16 (0.75)	−0.06 (0.75)	0.057	0.821
Blood analytes, mean (SD)							
Glucose (mg/dL)	1155	104.30 (11.84)	102.60 (11.70)	103.10 (12.45)	101.50 (10.87)	**0.033**	**0.009**
TG (mg/dL)	1154	77.90 (45.90)	80.58 (39.16)	77.21 (41.93)	70.43 (36.10)	**0.023**	**0.017**
TC (mg/dL)	1155	155.10 (33.63)	155.60 (34.06)	153.60 (36.76)	146.20 (32.59)	**0.003**	**0.002**
LDL-C (mg/dL)	893	78.57 (25.31)	79.11 (26.92)	79.26 (26.65)	74.60 (23.85)	0.196	0.131
HDL-C (mg/dL)	1153	63.35 (16.27)	63.18 (15.38)	63.45 (16.71)	61.41 (15.17)	0.367	0.185
Physical activity, mean (SD)							
Sedentary (min/day)	1121	608.49 (72.49)	604.05 (65.41)	609.12 (67.99)	601.94 (68.08)	0.539	0.426
MVPA (min/day)	1121	73.93 (23.88)	74.01 (22.81)	74.15 (23.17)	76.32 (23.80)	0.565	0.243
Step counts per day	1121	12,208.48 (2464.41)	11,977.63 (2375.30)	11,984.75 (2400.11)	12,243.93 (2436.11)	0.407	0.859
Tanner scale, mean (SD)	1188	3.11 (0.85)	3.28 (0.84)	3.17 (0.83)	3.08 (0.85)	**0.003**	0.365
Place of birth, *n* (%)						0.329	0.475
Spain	1082	268 (89.63)	268 (89.93)	275 (91.97)	271 (90.94)		
Rest of Europe	15	5 (1.67)	4 (1.34)	1 (0.33)	5 (1.68)		
Latin America	58	18 (6.02)	16 (5.37)	16 (5.35)	8 (2.68)		
Africa	10	1 (0.33)	4 (1.34)	1 (0.33)	4 (1.34)		
Other	29	7 (2.34)	6 (2.01)	6 (2.01)	10 (3.36)		
Education of mother, *n* (%)						0.052	0.285
Low	282	73 (26.55)	78 (28.78)	77 (28.31)	54 (20.53)		
Medium	429	108 (39.27)	95 (35.06)	119 (43.75)	107 (40.68)		
High	370	94 (34.18)	98 (36.16)	76 (27.94)	102 (38.78)		
Education of father, *n* (%)						0.490	0.345
Low	248	58 (28.57)	65 (31.40)	66 (32.20)	59 (29.21)		
Medium	330	91 (44.83)	86 (41.55)	80 (39.02)	73 (36.14)		
High	239	54 (26.60)	56 (27.05)	59 (28.78)	70 (34.65)		
Household income, *n* (%)						0.450	0.057
Low	366	93 (33.82)	103 (36.52)	89 (32.25)	81 (29.67)		
Medium	348	91 (33.09)	87 (30.85)	89 (32.25)	81 (29.67)		
High	392	91 (33.09)	92 (32.62)	98 (35.51)	111 (40.66)		
Municipality, *n* (%)						0.050	**0.006**
Barcelona	845	226 (75.59)	217 (72.82)	205 (68.56)	197 (66.11)		
Madrid	349	73 (24.41)	81 (27.188)	94 (31.44)	101 (33.89)		

TPE: Total polyphenol excretion in urine. GAE: Gallic acid equivalent. Q: Quartile of TPE. BMI: Body mass index (calculated as weight in kilograms divided by height in square meters). WC: Waist circumference. WtHR: Waist-to-height ratio. SBP: Systolic blood pressure. DBP: Diastolic blood pressure. TG: Triglycerides. TC: Total cholesterol. LDL: Low-density lipoprotein-cholesterol. HDL: High-density lipoprotein-cholesterol. MVPA: Moderate-to-vigorous physical activity. SFAs: Saturated fatty acids. MUFAs: Monounsaturated fatty acids. PUFAs: Polyunsaturated fatty acids. Statistical analyses were undertaken using one-way ANOVA for continuous variables and the chi-square (χ2) test for categorical variables. *p*-value refers to differences between quartiles of TPE. *p*-values < 0.05; values shown in bold are statistically significant.

**Table 3 antioxidants-09-00910-t003:** Mean dietary nutrients and food intake, according to TPE quartiles (mg GAE in urine/g creatinine).

TPE (mg GAE in Urine/g Creatinine)							
	*n*	Q1 (<71.8)	Q2 (71.9–111.1)	Q3 (111.2–161.2)	Q4 (>161.2)	*p-*Value	*p-*Trend
Dietary intake, mean (SD)							
Total energy (Kcal/d)	850	2423.92 (546.44)	2400.96 (578.92)	2482.93 (531.79)	2435.43 (527.03)	0.471	0.488
Protein (g/d)	850	115.01 (15.49)	116.31 (16.02)	117.43 (18.18)	117.09 (16.85)	0.455	0.150
Carbohydrates (g/d)	850	248.07 (34.03)	242.18 (36.36)	243.48 (35.43)	241.69 (35.89)	0.235	0.102
Fiber (g/d)	850	28.50 (7.05)	27.06 (6.90)	28.38 (7.05)	27.76 (7.01)	0.127	0.669
Total fat (g/d)	850	109.10 (13.50)	111.05 (14.11)	110.08 (14.55)	110.96 (14.37)	0.450	0.290
SFAs (g/d)	850	35.74 (5.92)	35.90 (5.32)	35.66 (5.43)	36.24 (5.99)	0.730	0.474
MUFAs (g/d)	850	50.11 (9.19)	49.34 (8.94)	50.32 (9.15)	50.09 (8.69)	0.691	0.738
PUFAs (g/d)	850	18.49 (3.65)	18.98 (4.04)	18.50 (4.20)	18.81 (4.06)	0.501	0.694
Food intake, mean (SD)							
Vegetables (g/d)	850	248.48 (124.88)	258.17 (123.87)	241.57 (119.68)	254.44 (127.15)	0.536	0.973
Fruits (g/d)	850	295.26 (162.15)	293.99 (171.00)	299.10 (167.52)	289.67 (153.63)	0.949	0.817
Legumes (g/d)	850	61.38 (31.98)	53.70 (28.25)	62.41 (31.02)	55.96 (31.15)	**0.007**	0.422
Cereals (g/d)	850	123.01 (54.37)	121.05 (55.71)	124.95 (55.80)	117.06 (50.62)	0.484	0.403
Dairy (g/d)	850	390.33 (189.09)	389.29 (192.34)	378.07 (185.58)	392.64 (187.61)	0.863	0.941
Meat (g/d)	850	217.57 (81.89)	215.50 (75.33)	223.94 (75.99)	213.27 (75.49)	0.526	0.852
Olive oil (g/d)	850	15.58 (10.58)	15.86 (12.30)	17.04 (14.14)	16.01 (11.67)	0.638	0.509
Fish (g/d)	850	80.60 (40.17)	79.03 (39.25)	84.97 (45.68)	80.63 (40.66)	0.494	0.634
Nuts (g/d)	850	9.31 (8.99)	9.66 (8.45)	9.67 (8.52)	8.52 (8.62)	0.482	0.379
Cookies, pastries, sweets, and snacks (g/d)	850	83,17 (42.73)	83.60 (41.57)	78.58 (39.93)	76.01 (42.50)	0.174	**0.039**
Chocolate (g/d)	850	6.71 (8.33)	7.32 (6.94)	6.33 (6.43)	7.68 (7.35)	0.230	0.392
Soft drinks (g/d)	850	140.80 (123.35)	133.78 (120.13)	136.91 (142.04)	134.27 (124.75)	0.939	0.675

TPE: Total polyphenol excretion in urine. GAE: Gallic acid equivalent. Q: Quartile of TPE. SFAs: Saturated fatty acids. MUFAs: Monounsaturated fatty acids. PUFAs: Polyunsaturated fatty acids. Statistical analyses were undertaken using one-way ANOVA for continuous variables. *p*-value refers to differences between quartiles of TPE. *p*-values < 0.05; values shown in bold are statistically significant.

**Table 4 antioxidants-09-00910-t004:** Association between body composition and BP with quartiles of TPE (mg GAE/g creatinine), according to gender.

		Total (*n* = 1194)						Girls (*n* = 569)				Boys (*n* = 625)		
		*n*	Q1 vs. Q4	*p-*Value	*p-*Trend	*p-*for Interaction	*n*	Q1 vs. Q4	*p-*Value	*p-*Trend	*n*	Q1 vs. Q4	*p-*Value	*p-*Trend
BMI z-score	Margin Mean		0.69 vs. 0.53					0.59 vs. 0.40				0.78 vs. 0.63		
	B (CI)—Model 1	1193	−0.16 (−0.40; 0.09)	0.204	0.072	0.144	569	−0.19 (−0.93; 0.55)	0.615	0.499	624	−0.15 (−0.29; −0.01)	**0.047**	**0.023**
	B (CI)—Model 2	999	−0.19 (−0.56; 0.19)	0.326	0.331	0.788	483	−0.26 (−0.97; 0.44)	0.466	0.448	516	−0.15 (−0.15; −0.14)	**<0.001**	**<0.001**
	B (CI)—Model 3	736	−0.25 (−0.79; 0.30)	0.379	0.446	0.548	352	−0.33 (−0.99; 0.33)	0.330	0.347	384	−0.17 (−0.50; 0.17)	0.335	0.543
WC z-score	Margin Mean		0.32 vs. 0.30					0.23 vs. 0.24				0.42 vs. 0.34		
	B (CI)—Model 1	1194	−0.03 (−0.30; 0.24)	0.846	0.578	0.297	569	0.01 (−0.64; 0.65)	0.986	0.903	625	−0.07 (−0.08; −0.07)	**<0.001**	**<0.001**
	B (CI)—Model 2	1000	−0.02 (−0.39; 0.35)	0.897	0.854	0.832	483	−0.03 (−0.52; 0.46)	0.910	0.892	517	−0.04 (−0.21; 0.13)	0.655	0.595
	B (CI)—Model 3	737	−0.06 (−0.54; 0.43)	0.812	0.827	0.613	352	−0.07 (−0.46; 0.33)	0.734	0.753	385	−0.03 (−0.50; 0.44)	0.903	0.941
Body fat %	Margin Mean		23.52 vs. 21.92					25.08 vs. 23.58				22.32 vs. 20.67		
	B (CI)—Model 1	1182	−1.59 (−3.61; 0.42)	0.120	**0.045**	**0.001**	566	−1.49 (−6.53; 3.53)	0.559	0.381	616	−1.65 (−2.64; −0.68)	**0.001**	**<0.001**
	B (CI)—Model 2	989	−1.58 (−3.75; 0.59)	0.153	0.143	0.923	480	−1.92 (−6.91; 3.05)	0.448	0.336	509	−1.58 (−2.72; −0.44)	**0.007**	**<0.001**
	B (CI)—Model 3	728	−1.94 (−4.66; 0.78)	0.162	0.213	0.867	349	−2.29 (−6.43;1.84)	0.277	0.226	379	−1.75 (−2.16; −1.36)	**<0.001**	**0.022**
SBP z-score	Margin Mean		−0.12 vs. −0.25					−0.13 vs. −0.24				−0.11 vs. −0.28		
	B (CI)—Model 1	1191	−0.14 (−0.27; −0)	**0.046**	**0.008**	0.465	567	−0.11 (−0.31; 0.09)	0.283	0.078	624	−0.16 (−0.26; −0.07)	**<0.001**	**<0.001**
	B (CI)—Model 2	999	−0.13 (−0.46; 0.20)	0.432	0.431	0.727	482	−0.13 (−0.32; 0.05)	0.156	**0.047**	517	−0.13 (−0.55; 0.30)	0.553	0.647
	B (CI)—Model 3	737	−0.16 (−0.49; 0.16)	0.329	0.434	0.280	352	−0.28 (−0.29; −0.28)	**<0.001**	**<0.001**	385	−0.07 (−0.55; 0.42)	0.780	0.959
DBP z-score	Margin Mean		−0.11 vs. −0.09					−0.04 vs. −0.01				−0.14 vs. −0.15		
	B (CI)—Model 1	1191	0.02 (0; 0.03)	**0.009**	0.136	**<0.001**	567	0.04 (0.02; 0.05)	**<0.001**	0.102	624	−0.01 (−0.02; 0)	0.173	0.708
	B (CI)—Model 2	1004	0.04 (0.01; 0.07)	**0.020**	**0.021**	0.670	482	0.02 (−0.05; 0.08)	**0.585**	0.611	517	0.04 (−0.08; 0.16)	0.511	0.376
	B (CI)—Model 3	737	−0.02 (−0.11; 0.08)	0.836	0.673	0.207	352	−0.07 (−0.12; −0.02)	**0.003**	0.059	385	0.04 (−0.09; 0.18)	0.545	0.361

TPE: Total polyphenol excretion in urine. Q: Quartile of TPE. GAE: Gallic acid equivalent. BP: Blood pressure. BMI: Body mass index. WC: Waist circumference. SBP: Systolic blood pressure. DBP: Diastolic blood pressure. B: Non-standardized coefficient. CI: Confidence interval. Model 1: unadjusted. Model 2: adjusted for gender (only for total participants), age, physical activity, fasting, Tanner stage, high parental education level, house income, and place of birth. Model 3: adjusted as in Model 2 plus energy intake, fiber, total fat, MUFAs, and PUFAs. *p*-value Q1 vs. Q4 of TPE, and *p*-trend < 0.05; values shown in bold are statistically significant.

**Table 5 antioxidants-09-00910-t005:** Association between blood analytes of cardiovascular health and quartiles of TPE (mg GAE/g creatinine), according to gender.

		Total (*n* = 1194)					Girls (*n* = 569)				Boys (*n* = 625)			
		*n*	Q1 vs. Q4	*p-*Value	*p-*Trend	*p-*for Interaction	*n*	Q1 vs. Q4	*p-*Value	*p-*Trend	*n*	Q1 vs. Q4	*p-*Value	*p-*Trend
Glucose (mg/dL)	Margin mean		103.05 vs. 100.66					101.76 vs. 99.55				105.09 vs. 101.54		
	B (CI)—Model 1	1155	−2.39 (−5.90; 1.12)	0.182	0.216	**<0.001**	546	−1.21 (−2.14; −0.27)	**0.012**	0.146	609	−2.57 (−9.74; 2.59)	0.256	0.233
	B (CI)—Model 2	1000	−2.66(−7.65, 2.33)	0.297	0.313	0.138	483	−1.54 (−4.31; 1.21)	0.272	0.482	517	−3.35 (−9.92; 3.23)	0.318	0.258
	B (CI)—Model 3	736	−1.76 (−6.24; 2.71)	0.440	0.491	**0.003**	352	2.40 (−0.71; 5.51)	0.130	**0.048**	384	−3.69 (−9.09; 1.71)	0.180	0.058
TG (mg/dL)	Margin mean		77.90 vs. 70.43					78.30 vs. 76.73				77.53 vs. 65.59		
	B (CI)—Model 1	1154	−7.47 (−9.11; −5.83)	**<0.001**	**<0.001**	**<0.001**	546	−1.57 (−5.53; 2.38)	0.435	0.260	608	−11.94 (−18.25; −5.62)	**<0.001**	**<0.001**
	B (CI)—Model 2	999	−7.41 (−16.11; 1.30)	0.096	**0.134**	0.354	483	−2.72 (−8.02; 2.59)	0.315	0.242	516	−10.18 (−23.27; 2.91)	0.127	0.182
	B (CI)—Model 3	735	−9.31 (−12.69; −5.15)	**<0.001**	**0.009**	**0.013**	352	4.40 (−4.81; 13.62)	0.349	0.577	383	−17.68 (−24.38; −10.99)	**<0.001**	**0.009**
TC (mg/dL)	Margin mean		149.29 vs. 142.42					149.05 vs. 145.99				149.61 vs. 139.87		
	B (CI)—Model 1	1155	−6.87 (−7.14; −6.60)	**<0.001**	**<0.001**	**0.048**	546	−3.07 (−3.91; −2.23)	**<0.001**	**0.001**	609	−9.74 (−10.78; −8.70)	**<0.001**	**<0.001**
	B (CI)—Model 2	1000	−6.54 (−8.06; −5.93)	**<0.001**	**<0.001**	**0.008**	483	−3.84 (−4.71; −2.97)	**<0.001**	**<0.001**	517	−8.55 (−9.01; −8.09)	**<0.001**	**<0.001**
	B (CI)—Model 3	736	−7.09 (−9.28; −4.90)	**<0.001**	**<0.001**	**0.017**	352	−3.11 (−4.19; −0.82)	0.060	**<0.001**	384	−8.66 (−14.23; −3.11)	**0.002**	**0.008**
LDL-C (mg/dL)	Margin mean		74.88 vs. 72.29					72.47 vs. 75.09				77.71 vs. 70.22		
	B (CI)—Model 1	893	−2.59 (−2.64; −2.55)	**<0.001**	**<0.001**	0.227	462	2.61 (0.98; 4.24)	**0.002**	**0.018**	431	−7.48 (−8.23; −6.75)	**<0.001**	**<0.001**
	B (CI)—Model 2	776	−2.83 (−5.81; 0.16)	0.063	0.057	**0.032**	408	1.82 (0.23; 3.42)	**0.025**	0.800	368	−7.53 (−11.35; −3.71)	**<0.001**	**0.007**
	B (CI)—Model 3	573	−1.98 (−4.09; −0.11)	**0.006**	**0.028**	0.177	300	−0.17 (−3.70; 4.05)	0.930	0.305	273	−4.09 (−9.75; 1.55)	**0.008**	**0.013**
HDL-C (mg/dL)	Margin mean		61.74 vs. 60.37					62.10 vs. 60.15				61.57 vs. 60.60		
	B (CI)—Model 1	1153	−1.37 (−1.82; −0.93)	**<0.001**	**<0.001**	0.183	546	−1.95 (−2.86; −1.04)	**<0.001**	**<0.001**	607	−0.96 (−1.12; −0.81)	**<0001**	**<0.001**
	B (CI)—Model 2	999	−1.39 (−3.25; 0.47)	0.142	0.249	0.563	483	−1.27 (−2.00; −0.53)	**0.001**	**<0.001**	516	−1.47 (−5.53; 2.59)	0.479	0.453
	B (CI)—Model 3	735	−1.88 (−4.92; 1.16)	0.227	0.272	0.685	352	−1.07 (−2.62; 0.47)	0.174	**<0.001**	383	−2.22 (−9.09; 4.64)	0.525	0.574

TPE: Total polyphenol excretion in urine. Q: Quartile of TPE. GAE: Gallic acid equivalent. TG: Triglycerides. TC: Total cholesterol. LDL-C: Low-density lipoprotein-cholesterol. HDL-C: High-density lipoprotein-cholesterol. B: Non-standardized coefficient. CI: Confidence interval. Model 1: unadjusted. Model 2: adjusted for gender (only for total participants), age, physical activity, fasting, Tanner stage, high parental education level, house income, and place of birth. Model 3: adjusted as in Model 2 plus BMI, WtHR, energy intake, fiber, total fat, MUFAs, and PUFAs. *p*-value Q1 vs. Q4 of TPE, and *p*-trend < 0.05; values shown in bold are statistically significant.

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
