# Peer review of "Polyphenols in Urine and Cardiovascular Risk Factors: A Cross-Sectional Analysis Reveals Gender Differences in Spanish Adolescents from the SI! Program"

_antioxidants, 2020, doi:10.3390/antiox9100910_

Round 1

Reviewer 1 Report

The paper has been improved much. In Table 4, the meaning of Q (Quartile of TPE?) should be described, for instance in footnote.

Author Response

The paper has been improved much. In Table 4, the meaning of Q (Quartile of TPE?) should be described, for instance in footnote.

Thank you for the kind comment. The meaning of Q (Quartile of TPE) has been described at the footnote of Tables 2, 3, 4, and 5.

Reviewer 2 Report

This is a revised manuscript related to polyphenols excreted (TPE) and the relationship to cardiovascular health predictors.  TPE have a relationship with some cardiovascular risk predictors in boys but not girls.  The authors have been somewhat responses to the previous concerns, and the results are fairly stated.  It is disappointing that the authors did not give a rationale for not analyzing TPE as a continuous variable.  Moreover, a potential reason for the sex dimorphism is unclear. Nevertheless, I have no further comments.

Author Response

This is a revised manuscript related to polyphenols excreted (TPE) and the relationship to cardiovascular health predictors.  TPE have a relationship with some cardiovascular risk predictors in boys but not girls.  The authors have been somewhat responses to the previous concerns, and the results are fairly stated.  It is disappointing that the authors did not give a rationale for not analyzing TPE as a continuous variable.  Moreover, a potential reason for the sex dimorphism is unclear. Nevertheless, I have no further comments

There is an interest in evaluating extreme groups, such as those with the highest and lowest quartile of TPE. Rank order data are often used in epidemiological studies to study the relationship between diet and disease. Data was also re-analyzed using structural equation modeling (SEM) to examine the relationship between TPE and CVRFs where both were considered as a continuous variable, however, the slopes were too low and barely explanatory.

We have considered to go deeper in the discussion about the influence of sexual dimorphism on the metabolism of polyphenols, in lines 395-407.

This manuscript is a resubmission of an earlier submission. The following is a list of the peer review reports and author responses from that submission.

Round 1

Reviewer 1 Report

The study included the interesting finding, while some points to be revised existed as follows:

1. Abstract: The authors should include the name of program as Title.

2. Introduction: The authors should summarize several polyphenol-related biomarkers (flavonoids, phenolic acids…some), in serum and urine, associated with cardiovascular risk factors.

3. Methods: The summary of characteristics of SI! Study should be briefly introduced and examined without citing the URL.

4. Measurements: The data of CV (accuracy and precision) and/ or detection limits of glucose, lipids and TPE were described.

5. Results: Table 2 and 3: The authors should show the data as ‘standardized’ beta.

6. Results: Table 2: The results of girls in BMI-Z should be shown by the second decimal place.

7. Discussion: The authors should more explain and speculate the mechanistical reasons why some lipids, ‘not blood pressure and glucose’, were correlated to TPE.

8. Discussion: As for the intake, serum concentration and urine levels, the correlations and turn speeds should be included and discussed.

9. Line/row 264: ‘That’ was necessary between…possible and boys….

10. Line/row 268-71: In this sentence, the conjunction word was needed.

Author Response

The study included the interesting finding, while some points to be revised existed as follows:

  1. Abstract: The authors should include the name of program as Title.

We have considered this recommendation in the abstract (line 42).

  1. Introduction: The authors should summarize several polyphenol-related biomarkers (flavonoids, phenolic acids…some), in serum and urine, associated with cardiovascular risk factors.

This has been done in the Introduction as suggested (lines 82-94).

  1. Methods: The summary of characteristics of SI! Study should be briefly introduced and examined without citing the URL.

We have summarized the characteristics of the SI! Study in Methods (lines 101-112).

  1. Measurements: The data of CV (accuracy and precision) and/ or detection limits of glucose, lipids and TPE were described.

We have described the accuracy, precision and detection limit of glucose, lipid profile, and TPE according to a validated method (Lines 139-143, 150-158) and the method has been cited with the measurements.

  1. Results: Table 3and 4: The authors should show the data as ‘standardized’ beta.

Unstandardized coefficients are useful for interpreting our regression results. The value of any unstandardized coefficient denotes the change in the dependent variable with a unit increment in the independent variable.

  1. Results: Table 3: The results of girls in BMI-Z should be shown by the second decimal place.

This has now been done.

  1. Discussion: The authors should more explain and speculate the mechanistical reasons why some lipids, ‘not blood pressure and glucose’, were correlated to TPE.

We have explained and speculated about the mechanistic reasons why some lipids and not blood pressure and glucose were correlated with TPE in the Discussion (line 352-371).

  1. Discussion: As for the intake, serum concentration and urine levels, the correlations and turn speeds should be included and discussed.

We considered intake and inter-individual variability factors in the statistical analysis and discussion.

  1. Line/row 264: ‘That’ was necessary between…possible and boys….

“That” has been omitted.

  1. Line/row 268-71: In this sentence, the conjunction word was needed.

A conjunction has been included.

Reviewer 2 Report

This manuscript analyzes total polyphenols excreted (TPE) in relation to cardiovascular health predictors.  It finds a sexual dimorphism where TPE are inversely related to cardiovascular risk predictors in adolescent boys but not girls.

The study appears carefully done. There are a number of considerations that could be addressed to strengthen the conclusions and provide a better context for readers:

  1. Polyphenols have been measured in several populations.  It would be nice to see if the results hold in an adult population and if the sexual dimorphism is still present.
  2. It would be desirable to state the puberty status of the study participants to make sure that this is not an issue.  If most are prepuberty, it would be nice to have some hypothesis about why the boys are different from the girls.
  3. Please state the coefficients of variation between measures and between patients.  TPE tends to have wide variability, making conclusions more difficult.
  4. If treated as a continuous variable is TPE associated with cardiovascular risk?  What was the justification for quartiles?  This might establish a dose response that would add more weight to the relationship of TPE and cardiovascular risk.
  5. The assumption is that TPE is proportional to dietary intake.  This does not appear to be the case in this study.  Does this mean that gut production, gut transport, and renal excretion are changing?  Is anything known about differences in these parameters by gender?
  6. Do boys simple have more total intake of food since they are larger and, therefore, do they have a greater TPE?
  7.  

Author Response

This manuscript analyzes total polyphenols excreted (TPE) in relation to cardiovascular health predictors. It finds a sexual dimorphism where TPE are inversely related to cardiovascular risk predictors in adolescent boys but not girls.

The study appears carefully done. There are a number of considerations that could be addressed to strengthen the conclusions and provide a better context for readers:

1. Polyphenols have been measured in several populations. It would be nice to see if the results hold in an adult population and if the sexual dimorphism is still present.
2. It would be desirable to state the puberty status of the study participants to make sure that this is not an issue. If most are prepuberty, it would be nice to have some hypothesis about why the boys are different from the girls.
Tanner stage was considered a covariate in the regression models.
3. Please state the coefficients of variation between measures and between patients. TPE tends to have wide variability, making conclusions more difficult.
The CVs have been stated between each TPE measure in Methods (Iine 150-158) and the TPE among patients in Results (line 221).
4. If treated as a continuous variable is TPE associated with cardiovascular risk? What was the justification for quartiles? This might establish a dose response that would add more weight to the relationship of TPE and cardiovascular risk.
We categorized TPE into quartiles because it is a common practice in epidemiological studies. However, the analysis was performed considered TPE as a continuous variable and the p-values did not differ (data not shown).
5. The assumption is that TPE is proportional to dietary intake. This does not appear to be the case in this study. Does this mean that gut production, gut transport, and renal excretion are changing? Is anything known about differences in these parameters by gender?
We do not have information about gut production or gut transport. However, in this analysis, we selected adolescents without chronic diseases such as diabetes and hypertension according to the general questionnaire described. In addition, we excluded participants who had taken drugs or dietary supplements in the days prior to the analysis. We determined urinary creatinine concentrations, and in the absence of disease, creatinine in urine can be used to determine urinary excretion of compounds in spot urine samples. For this reason, TPE has been expressed as mg gallic acid equivalent (GAE)/g creatinine.
6. Do boys simple have more total intake of food since they are larger and, therefore, do they have a greater TPE?
We considered this observation in the linear regression models adding total energy, fat, MUFAs, PUFAs, and fiber intake as covariates.

Reviewer 3 Report

I believe that the current manuscript describes a focused analysis that aligns well with the research question in-hand. The findings presented should be of broad interest to the readership of "Antioxidants". The following comments are aimed at improving the clarity and appropriateness of the current manuscript:

  1. Lines 165-173 - I suggest moving this section elsewhere. It seems odd to start with a description of the supplementary data rather than the findings presented in the manuscript itself. I suggest also including summary data (means, SDs and p-values or similar) within the text to help support the statements made.
  2. Discussion - line 259 onwards - the authors have highlighted the potential for differences in males and females in terms of polyphenol absorption in the small intestine and then make vague mention of  the gut microbiome but have not presented information as to whether dietary intake is different in males and females of this age. This section needs to be expanded to consider whether dietary habit could be a major driver of the outcomes noted and to provide a more substantiated argument to support the potential for differences in the microbiota of males and females (of this age ideally) and how this might impact on appearance and phenolic compounds in the bloodstream and subsequently the urine. The potential that urinary output could also impact on the major outcome measure (concentration of GAE in the urine has also not been considered).

Author Response

I believe that the current manuscript describes a focused analysis that aligns well with the research question in-hand. The findings presented should be of broad interest to the readership of "Antioxidants". The following comments are aimed at improving the clarity and appropriateness of the current manuscript:

1. Lines 165-173 - I suggest moving this section elsewhere. It seems odd to start with a description of the  supplementary data rather than the findings presented in the manuscript itself. I suggest also including summary data (means, SDs and p-values or similar) within the text to help support the statements made.
This has been changed as suggested and has been moved to Results.
2. Discussion - line 259 onwards - the authors have highlighted the potential for differences in males and females in terms of polyphenol absorption in the small intestine and then make vague mention of the gut microbiome but have not
presented information as to whether dietary intake is different in males and females of this age. This section needs to be expanded to consider whether dietary habit could be a major driver of the outcomes noted and to provide a more substantiated argument to support the potential for differences in the microbiota of males and females (of this age ideally) and how this might impact on appearance and phenolic compounds in the bloodstream and subsequently the urine. The potential that urinary output could also impact on the major outcome measure (concentration of
GAE in the urine has also not been considered).
We considered this observation in the analysis separating by sex and in the linear regression models by adding total energy, fat, MUFAs, PUFAs, and fiber intake as covariates. On another hand, we considered dietary intake in discussion.

Reviewer 4 Report

A cross-sectional study was performed in the present study in a cohort of 1326 Spanish adolescents. Anthropometric parameters together with blood pressure, plasma glucose and lipid profile. The major issue is to investigate “total polyphenol excretion in urine” which was determined by the Folin-Ciocalteu method and is worthy to study it. However, the association between “total polyphenol excretion in urine” and other comment parameter in blood pressure, plasma glucose and lipid profile are too weak to answer the relation among “total polyphenol excretion in urine”, blood pressure, plasma glucose and lipid profile. Polyphenol is the most important anti-oxidative nutrients. Therefore, missing oxidative biomarkers from blood sampling were the biggest weakness of the present study. Since the present study investigate “total polyphenol excretion in urine”, some parameters from blood sampling or re-checking normal renal function such oxidative status, inflammatory markers, or renal markers (urine) should be included in the present study. The study design is suffered from several missing important parameters.

In methodological issues, the poor data analysis of the present study is suffered from only using lineal regression models which can not tell the relations of “total polyphenol excretion in urine” under multiple interaction. For example, it is common using Structural equation modeling (SEM) to observe multiple parameters to influence “total polyphenol excretion in urine”. Besides, cardiovascular health appeared to be over-interpreted because the present study only show BMI, blood pressure, plasma glucose and lipid profile. Cardiometabolic risk factors may be suitable for the present study. Major conclusion “ Higher concentrations of “total polyphenol excretion in urine” were associated with a better profile of cardiovascular health in male adolescents.” is too weak to be a novel finding and did not answer an important question to solve a clinical problem.

Author Response

A cross-sectional study was performed in the present study in a cohort of 1326 Spanish adolescents. Anthropometric parameters together with blood pressure, plasma glucose and lipid profile. The major issue is to investigate “total polyphenol excretion in urine” which was determined by the Folin-Ciocalteu method and is worthy to study it. However, the association between “total polyphenol excretion in urine” and other comment parameter in blood pressure, plasma glucose and lipid profile are too weak to answer the relation among “total polyphenol excretion in urine”, blood pressure, plasma glucose and lipid profile. Polyphenol is the most important anti-oxidative nutrients. Therefore, missing oxidative biomarkers from blood sampling were the biggest weakness of the present study. Since the present study investigate “total polyphenol excretion in urine”, some parameters from blood sampling or rechecking normal renal function such oxidative status, inflammatory markers, or renal markers (urine) should be included in the present study. The study design is suffered from several missing important parameters.

-We aimed to evaluate the relationship between polyphenol excretion in urine and cardiovascular risk factors, in this case, oxidative biomarkers were not considered in the analysis. On the other hand, in the original protocol of the Si! Program for secondary schools blood samples extraction was not considered because this program was aimed at evaluating cardiovascular health parameters with analysis using non-invasive techniques such as urine samples. We only determined the lipid profile and glucose levels using a rapid and simultaneous analysis by cardiocheck
plus, using 40 ul of blood from one fingerstick.
- For this analysis we selected adolescents without chronic diseases such as diabetes according to the general questionnaire described. In addition, we excluded participants who had taken drugs or dietary supplements in the day prior to the analysis. In the absence of reported disease, creatinine concentrations in urine are usually very stable and can be used to estimate the urinary excretion of substances with only spot urine samples. For this reason, TPE was expressed as mg gallic acid equivalent (GAE)/g creatinine.

In methodological issues, the poor data analysis of the present study is suffered from only using lineal regression models which cannot tell the relations of “total polyphenol excretion in urine” under multiple interaction. For example, it is common using Structural equation modeling (SEM) to observe multiple parameters to influence “total polyphenol excretion in urine”. Besides, cardiovascular health appeared to be over-interpreted because the present study only shows BMI, blood pressure, plasma glucose and lipid profile. Cardiometabolic risk factors may be suitable for the present study. Major conclusion “ Higher concentrations of “total polyphenol excretion in urine” were associated with a better profile of cardiovascular health in male adolescents.” is too weak to be a novel finding and did not answer an important question to solve a clinical problem.

- We have taken this recommendation into account using structural equation modeling (SEM) to observe multiple parameters that influence “total polyphenol excretion in urine” following a factorial analysis (Line 204-213, 281-292). In our hypothesized model gender, age, physical activity, fasting, tanner scale, energy intake, fat intake, MUFAs, PUFAs, fiber intake, high parental education, place of birth, and house income were considered as covariates.
- In the title, we have changed “cardiovascular health indicators” to “cardiovascular risk factors” as suggested.
- Concerning conclusions, we only could report the relationship found between polyphenols in urine and cardiovascular risk factors, because this was a crosssectional analysis. For this reason, in the Discussion, we have included the limitation of the design of the analysis for determining casual-effect.